# Theoretical and Experimental Investigation of Bonded Patch Repairs of a Rubber Reinforced Composite Conveyor Belt

**DOI:** 10.3390/polym13111710

**Published:** 2021-05-24

**Authors:** Rawdha Kessentini, Olga Klinkova, Imad Tawfiq, Mohamed Haddar

**Affiliations:** 1Laboratoire de Mécanique Modélisation et Productique (LA2MP) LR13ES25, Ecole Nationale d’Ingénieurs de Sfax, BP N° 1173, Sfax 3038, Tunisia; mohamed.haddar@enis.rnu.tn; 2Laboratoire QUARTZ (EA7393), ISAE-Supméca, 3 rue Fernand Hainaut, Saint Ouen, CEDEX, 93407 Paris, France; olga.klinkova@supmeca.fr (O.K.); imad.tawfiq@supmeca.fr (I.T.)

**Keywords:** rubber, textile reinforcement, experimental results, bidirectional linear analysis, hygro-thermo mechanical loading, bonding assembly

## Abstract

The present study proposes a reparation method for designing and optimizing a rubber to rubber and rubber to textile reinforcement. The present application is the conveyor belt used in the transport industry. The tensile behavior of the repaired specimens was studied using experimental results. A bidirectional linear analysis allows us to predict the effect of geometric parameters on the stress concentration zone of the repaired belt under hygro-thermo mechanical loading and its consequence on the integrity of the structure. A tensile test was carried out in order to investigate the behavior of a repaired specimen made with a rubber cover patch and an inner composite patch. Two stacking sequences of an inner composite patch and the material properties are considered in the parametric study in order to reduce the stress concentration in the parent belt. The correlation between the theoretical and experimental results allows us to define a strength tool to understand the load transfer from rubber to a textile rubber patch.

## 1. Introduction

Conveyor belts are widely employed in the transportation of abrasive materials such as coal from mines, storage depots or other sources of particulate materials. The belt is the sensitive component of the conveyor; it is a rubber composite reinforced by a textile carcass and covered with a supplementary protection rubber ply at the upper and bottom sides of the belt. The belt surface has a load-carrying function; therefore, it is subjected to wear during service. Extreme operating conditions can cause local damage of the composite conveyor belt and reduce mechanical performance. Frequently, the cost of a systematic replacement of a damaged belt is too high [1,2]. In order to stop the propagation of local damages and cracks, a local repair can be considered to be a good solution for economical and mechanical reasons. One of the repair methods used by the transportation industry consists of composite patches bonding the damaged zone. Composite patch reinforced technology is actually dominated by applications in aluminum aircraft structures followed by a civil engineering structure. The design of a repair structure requires that the patch should absorb the load imposed during the service operation. The experimental study of Yala and Megueni [3] focused on the geometric parameters of patches and adhesives; they investigated the effects of the patch and adhesive thicknesses and the shear modulus of the adhesive upon the stress intensity factor. Later, Khalili et al. [4] dealt with the experimental study of the mechanical behavior of an edge-notched steel plate repaired by a polymer composite and smart patches. The polymer composite patches were made of Kevlar/epoxy and glass/epoxy or a combination of both. They proved that the patching had a significant effect on the mechanical properties of the notched plate; more than that, the best patches subjected to tension, bending and impact loading were reinforced with nickel/titanium and the smart patches had a more significant effect on the above behavior compared with the composite patches. Errouane et al. [5] developed a numerical model for the optimization of a composite patch repair of an aluminum plate containing a central crack under a tension load. Their optimization model proved that the eccentricity reduced with the decrease of the patch thickness during load transfer in the repaired region. They also increased the efficiency of composite patches with less volume of materials leading to a significant lifespan increase of aluminum cracked plates according to service conditions. Tran et al. [6] et Serrano-Garcia at al. [7] are developed embedded the nanofibers into textiles to make them active textiles.

Up to present, much attention has been paid to rubber to rubber and rubber to textile bonding applications especially in repair use. The aim of this study is to develop a technical tool for designing and optimizing a rubber to rubber and rubber to textile reinforcement method for the reparation of a belt structure. This is achieved through a combined approach consisting of experimental characterization and an analytical bidirectional hygro-thermomechanical analysis. In the first part of experimental works, the tensile behaviors of repaired specimens with two series patches, one with a stacking sequence in the yarn direction and the second with a stacking sequence in the warp direction of the textile ply, were investigated. This work allowed us to highlight the influence of the stacking sequence on the repair performance. In the second part of experimental works, the mechanical parameters of a healthy belt, textile carcass and cover rubber layer are determined in order to have material data of the bidirectional hygro-thermomechanical model. The stress distribution in a 2D plane is analyzed for the parent belt, inner textile patch and rubber cover patch of the repaired structure. The correlation between the analytical and experimental results allows us to define a strength tool to understand the load transfer from rubber to a textile rubber patch by using the bidirectional stress distribution in a bonded patch repair of the conveyor belt. Finally, a dimensionless design is proposed by means of a parametric study that consists of the determination of the influence of individual parameters on the repair performance.

## 2. Experimental Method

### 2.1. Material

A rubber-textile conveyor belt represented in Figure 1 was repaired according to the structural reparation method. The fabric textile carcass was designed to be layered with four plies of polyester (PET) yarn in the warp direction and polyamide (PA6) yarn in the weft direction.

During conveyor belt operation, damage can occur. Due to the heterogeneity of the structural composite belt, four major damage mechanisms were identified and can be seen in Figure 2: (1) matrix cracking (at a microscopic scale), (2) delamination (at a meso-scale), (3) breaking and buckling of fibers (at a microscale) and finally (4) fiber-matrix interfacial decohesion (at a microscale). This damage presented the critical case when the damage occurred at the half of the conveyor belt thickness. The reparation of such a rubber reinforced composite will be described in the next section.

### 2.2. Repaired Specimen Manufacturing

Two series of five repaired specimens were prepared in a load and a transverse direction, as shown in Figure 3. The dimensions of all specimens are shown in Figure 4.

The represented diagram of the bonded patch repair technology of the specimens is shown in Figure 5. The 80 mm length across 6 mm thickness in the parent specimen was to simulate the damage zone that had been cleaned up. The damage had to be removed. While roughness plays an important role in adhesion phenomena, the surface topography of the peeling and milling damage removing processes was measured by surface scan means, as shown in Figure 6.

The peeling was compared with the milling techniques in terms of surface roughness, surficial fiber damage, process rapidity and quantity of dust generated. With respect to the roughness results obtained (Figure 6) and with the above-described parameters, the upper rectangular covered patch was cut from the belt material using the peel technique (Figure 7a). The inner patch was prepared from textile ply made with the same plies as the carcass fabric of the belt (Figure 7b). Two sequences of inner patches [0]_2_ and [90]_2_ were prepared to repair the specimens in the longitudinal and the transverse directions, respectively, as shown in Figure 7b.

In order to bond the repair system composed from the parent belt, inner patch and covered patch, two different processes were used: hot vulcanization and cold polymerization. The cold polymerization process makes it possible to obtain the adhesion of rubber-rubber surfaces by means of the ELASTOGLUE 2000 adhesive without any vulcanization equipment. The adhesive used is applicable up to 3 h after its activation by the curing agent. It was prepared and thoroughly mixed in the portion of 1 kg ELASTOLOGUE 2000 adhesive with 60 g of hardener. During the phase of overlapping the inner and covered patches, it was necessary to ensure the coincidence between the textile plies of the inner patch and the parent belt in each layer. For this reason, a roller was used to avoid the porosities between layers. Pressure had then to be applied on the repair system with a rubber hammer to expel the air. Maximum polymerization was obtained after 12 h at 23 °C under three bars and 40% of relative humidity.

The hot vulcanization process activates strong links and creates bridging points between rubber-textile fabric plies of the inner patch and the rubber-covered patch. It is able to create covalent bonds between macromolecular chains to prevent chain sliding and give rubber its elastic behavior. After overlapping the rubber layer, the inner layer and the covered patches by the same technique as cold polymerization by using ELASTO-DISOL adhesive, a hot vulcanization cycle was applied by means of an electric DK 20T press. The vulcanization cycle information can be found in [8].

### 2.3. Adhesive Specimen Manufacturing

To be able to characterize the ELASTOGLUE 2000, the adhesive tensile specimens were cut from a molded plate of 250 × 250 mm^2^, which was molded using a two-mold set-up covered with a Teflon coating. The adhesive was prepared and thoroughly mixed in the portion mentioned before (1 kg/60 g), then placed in the mold using a spatula. Finally, the mold was closed and then inserted into a climatic chamber to stabilize the temperature (23 °C) and humidity (40%) environment. After three days, the plaque was polymerized. Three adhesive specimens were cut following the dimensions presented in Figure 8.

### 2.4. Mechanical Test

In the first part of the mechanical characterization, static tensile testing was conducted on a healthy belt and a textile carcass in both the load and the transverse directions. The tests were carried out in accordance with the ISO283: 2015 [9] standard. Five specimens in each direction were cut in full thickness of the belt as shown in Figure 3. The other three textile carcass specimens were obtained after the elimination of the top and bottom cover layers by using a peel technique. The testing was performed using an Instron tensile test machine equipped with transverse ridge jaws in order to prevent specimens from sliding during the test. The useful dimensions 100 × 25 mm^2^ were fixed in the center of the specimen. The test rate was 100 mm/min. A tensile test was also carried out for the top cover rubber through the same test conditions. Three samples were obtained from the whole belt specimens after the extraction of the textile carcass samples. The tensile tests were run at room temperature.

Furthermore, a tensile test was conducted for the ELASTOGLUE 2000 adhesive specimens with normalized dimensions [10] as described in Figure 8 with a loading speed of 2 mm/min.

Static testing was then conducted on repaired specimens with two patch series, one with the stacking sequence in the yarn direction and the second with the stacking sequence in the warp direction of the textile ply. Five specimens were tested in each configuration in order to understand the impact of the fiber orientation on the repair behavior. The same test conditions as the first characterization part were used.

## 3. Bidirectional Hygro-Thermomechanical Modelling

The bonded patch repair of the conveyor belt was composed of the parent belt, an inner patch and a covered patch, as shown in Figure 9. The bidirectional model detailed in [11] was used in order to determine the stress distribution in the patches and the parent belt. A previous study had shown that this model was well suited to the analysis of composite materials [11]. The belt had an elastic-visco-plastic behavior but only the elastic domain was considered in this analysis. The adhesive had a hyper-elastic behavior and it was modeled by the Ogden model (N = 3) as demonstrated in Figure 10. More details about the experimental identification and a comparison of the analytical models can be found in Kessentini et al. [12]. The patches, parent belt and adhesive were assumed to be linear elastic and their mechanical characteristics did not depend on temperature, humidity and time. Shear deformation existed in the adhesive interfaces while normal stresses rose in the parent belt, inner patch and covered patch. A perfect adhesion was considered in both interfaces of the bonded joints. The mechanical properties are presented in Table 1 where E and ν refer to Young’s modulus and Poisson’s ratio, respectively. *R* refers to the elastic resistance. Subscripts x, y, z correspond with the longitudinal direction and the two transverse directions. The length, width and thickness were denoted Lx, Ly and h, respectively as listed in Table 2. The thickness of the parent belt, the thickness of the inner patch, the thickness of the covered patch and the thickness of the adhesive layer are denoted *h*_1_, *h*_2_, *h*_3_ and *e_c_*.

The calculations of the stress distribution in the repaired structure were carried out considering the hygro-thermomechanical coupling effects. The coefficients of the thermal expansion (CTE) of the three layers were the CTE of polyester, polyamide and that of rubber as listed in Table 3. The homogenization theory was then applied to estimate the CTE of the composite belt. It was assumed that the belt had the same CTE as the textile carcass in the longitudinal and the transverse directions. To achieve the thermal effect, a temperature gradient of 50 °C was taken into account in this analysis.

The effect of moisture diffusion was modelled by means of the moisture concentration induced in the covered patch as described in Figure 9. As demonstrated in the last study [12], the effect of moisture diffusion of this material was achieved by using a sequential dual Fickian (SDF) model. The moisture concentration was determined at any time *t* at each spatial coordinate *Z* by Equation (1):(1)$$\begin{array}{r} ∆C\left(z,t\right)=\left[1-\frac{4}{\pi }\overset{\infty }{\underset{N=1}{\sum }}\frac{{\left(-1\right)}^{N}}{\left(2N+1\right)}exp\left[-\frac{{\left(2N+1\right)}^{2}\pi^{2}D_{1}t}{h^{2}}\right]cos\frac{\left(2N+1\right)\pi Z}{h}\right]\times \frac{C_{1\infty }}{C_{\infty }}+\phi \left(t-t_{d}\right)\\ \times \left[1+\frac{4}{\pi }\overset{\infty }{\underset{N=1}{\sum }}\frac{{\left(-1\right)}^{N}}{\left(2N+1\right)}exp\left[-\frac{{\left(2N+1\right)}^{2}\pi^{2}D_{2}\left(t-t_{d}\right)}{h^{2}}\right]cos\frac{\left(2N+1\right)\pi Z}{h}\right]\times \frac{C_{2\infty }}{C_{\infty }} \end{array}$$
where *C*_1∞_ and *C*_2∞_ were the saturated concentrations of the first and second diffusion mechanisms and *C*_1∞_ + *C*_2∞_ = *C*_∞_, with *C*_∞_ as the total saturation concentration of the whole process. *D*_1_ and *D*_2_ were the diffusion coefficients of the first and second diffusion mechanisms, respectively. *t_d_* was the retarding time of the second moisture uptake mechanism at which the transition from the first moisture diffusion process to the second one occurred and ϕ(*t*) was the Heaviside step function. The diffusion parameters are listed in Table 4. The hygroscopic stress distribution field was determined by the product of the concentration Δ*C* and the hygric expansion coefficients (CHE) of the parent belt, inner patch and cover patch outputted in Table 5.

Hygro-thermomechanical stress distributions were calculated when the repaired structure was exposed to a mechanical tensile load of σ_xx_ = 2 MPa, temperature gradient of ΔT = 50 °C and a moisture concentration of Δ*C* = 0.4%. This value was determined experimentally [12] to be reached after four months of water immersion at 50 °C.

## 4. Results and Discussion

### 4.1. Experiment Test Results

Experimental tests of the healthy belt and the bonded repaired specimens were carried out with two sequences of the inner patch. Figure 11 shows the stress-strain curves for the transverse and longitudinal repaired specimens using a polymerization bonding process compared with healthy belt specimens. Series (L) were designed to use [0]_2_ inner patches. For series (T), patches [90]_2_ were used to repair the specimen in the transverse direction. The fiber orientation of the patch was the same as those specimens’ directions. The results are resumed in Table 6 where $E^{rep}$ and $R^{rep}$ referred to the Young’s modulus and failure resistance of the repaired specimen. $E^{ref}$ and $R^{ref}$ were the Young’s modulus and the failure resistance of the healthy specimen. The following statements of experimental research were:The stress-strain curves obtained on the two sets of repaired specimens by the polymerization process presented the same behavior as those on the healthy specimens. However, a loss of tensile strength in the repaired structure was estimated at 57% for series (L) and at 39% for series (T) as listed in Table 6.An attenuation of the director coefficient in the elastic domain was observed. Consequently, the reduction of the Young’s modulus of the repaired structure was estimated at 20% for series (L) and at 13% for series (T).On the curves, an important decrease of the director coefficient in the elastic and plastic domains was observed for series (L) of the repaired specimens because of the concentration of high stresses located in the joint’s interfaces, which favored the important damage in the repaired specimens and led to premature failure.Series (T) repairs using the [90]_2_ stacking sequence have the best performance in terms of resistance, which was up to 68.73% (Table 6) while repairs of series (L) by means of the most rigid patch [0]_2_ seemed the least effective. It was then observed that the performance of the repairs by the polymerization process decreased when the rigidity of the plies and test specimen increased. The repair was not perfect if the inner patch was too soft or too rigid. Therefore, optimization of the inner patch was necessary to achieve the best repair performance.For series (T), the cover patch was still attached to the parent belt as shown in Figure 11 even if the fibers were broken and completely separated from the parent belt. This implied that the cover patch in the parent belt, which was adjacent to the top adhesive layer, was particularly not damaged and the hyper-elastic behavior led to the absorption of the transfer shear stress. The observation of the covered patch in domain D proved that the orientation of the specimens relative to the stacking sequence and the rigidity of the inner patch had a significant effect on the rupture mechanism and the mode of repaired specimen damage.

### 4.2. Bidirectional Analysis

#### 4.2.1. Shear Stress Distribution

Making the repaired structure more durable under external loading was the objective of this study. In fact, joints are the most critical elements in a bonded structure; they assume the load transfer mechanism between the adhesive and the adherents. Figure 12 shows the shear stress distribution in both interfaces of the repaired belt structure. τ_ixz_ and τ_iyz_ referred to the shear stress in the XZ and YZ planes of the interface (1) i = 1 between the parent belt and the inner patch and for the interface (2) between the inner patch and the covered patch, i = 2. It was found that the maximum shear stress was always in the interface (2). This was due to the residual stresses generated in the covered patch as a result of the moisture diffusion. The covered patch was the most sensitive component to water expansion so the strain fields accumulated at the interface between the covered patch and the inner patches. The shear stresses in the XZ plane were the most important; this was the direction of the mechanical load. The most stressed zone in both joints was located near the end edges.

#### 4.2.2. Normal Stress Distribution

The bidirectional model was carried out by applying a uniform tensile stress of σ_xx_ = 2 MPa at both ends of the repaired belt structure, which remained within the elastic domain of the composite material of the repaired belt structure. The tensile load was coupled with the temperature field of ΔT = 50 °C and the moisture diffusion ΔC = 0.4% on the covered patch.

Figure 13 shows the normal distribution generated on the parent belt, inner patch and covered patch under hygro-thermomechanical loads. It was found that the inner patch absorbed approximately twice the maximum stresses generated in the parent belt so that the contribution of the composite patch therefore played an important role in the equilibrium mechanism of the stress distribution in the repaired structure. A tension distribution was defined in both the parent belt and the inner patch while compressive distribution was seen in the covered patch, which was due to the residual stress generated from the coupled effect on the covered patch subjected to moisture diffusion and a thermal field. According to the experiment results, normal stress in the covered patch was equal to zero near the width end so that the cover patch was still attached to the parent belt, as seen in zone D of Figure 11.

### 4.3. Parametric Study

A repair will be efficient and effective when each component is chosen so that the distribution of normal and shear stresses is evenly spread and the maximum value in the repair system defined as normalized normal σ* and normalized shear τ* repair is at minimal. In Figure 13, the stress concentrations were situated at the edges of the repaired specimen, which was in accordance with Gong et al. [13]. When the composite and the repairing patch was assembled by bonding onto the specimen, the stresses were redistributed between the patch and the parent composite so a new zone of stress concentration might be located. According to the location of stress concentration, the critical zones in a patch repaired system may occur in six possible locations, as indicated in Figure 12 and Figure 13:Zone 1: at the center of the parent belt.Zone 2: near the transverse edges of the inner patch.Zone 3: near the longitudinal edges of the inner patch.Zone 4: at the longitudinal edges of the covered patch.Zone 5: at the longitudinal edges of the interface between the inner and cover patches.Zone 6: at the transverse edges of the interface between the inner and cover patches.

#### 4.3.1. Influence of Young’s Modulus of the Adhesive E_a_

The effect of Young’s modulus of the adhesive, E_a_, on the performance of a patched repair was studied by varying E_a_ from 0.05 to 7 MPa while the thickness of the patches, the parent belt and of the adhesive joint was constant. The results are plotted in Figure 14 and Figure 15. $\tau_{max}^{*}$ is the normalized shear stresses obtained by the square between the shear stresses and the maximum shear stress obtained in the joint adjacent to the cover patch for a reparation with composite patches at 0°. As shown, the maximum shear in the XZ and YZ plane depended on the adhesive Young’s modulus. Increasing the E_a_ value resulted in an important increase in $\tau_{max}^{*}$ of the second interface. The maximum normal stress $\sigma_{max}^{*}$ in the patches and the parent belt was still invariant with the variation of the adhesive Young’s modulus.

#### 4.3.2. Influence of the Thickness Adhesive *h_a_*

The effect of adhesive thickness was studied by keeping a constant adhesive modulus (E_a_ = 7 MPa) and constant patch and parent belt thicknesses. In practice, the variation of the adhesive thickness was confined to a range between 0.01 and 1 mm. As shown in Figure 16 and Figure 17, both $\tau_{max}^{*}$ and $\sigma_{max}^{*}$ of the inner patch and the parent belt depended on the adhesive thickness; keeping *h_a_* between 0.01 and 1 mm led to an increase of the $\tau_{max}^{*}$ in both interfaces. In the range 0.03–0.13 of varying *h_a_*, both interfaces in the YZ plane had the same $\tau_{max}^{*}$. It could be deduced that the intersection of these two curves would give the optimum system performance. The values of *h_a_* at the points shown in Figure 16 corresponding with the value of $\sigma_{max}^{*}$ in interface (1) and equal to $\sigma_{max}^{*}$ in interface (2) were defined as the optimal adhesive thickness for the repaired system.

#### 4.3.3. Influence of Young’s Modulus of the Composite Patch

To better understand the role of the inner patch made with two textile plies and three rubber layers on the behavior of the repair belt system, the sensitivity of the rigidity of composite patch was investigated. Within one repair, the fiber orientation was uniform plies. However, the orientations of each layer in the repair could change independently of the other repair layers. In this section, the influence of the rigidity of the polyester (PET) fibers was studied by keeping a constant composite patch modulus in a transverse direction ($E_{2y}^{ref}$ = 46.54 MPa) and a constant cover patch and parent belt Young’s modulus. The influence of the rigidity of polyamide (PA6) fibers was studied by keeping a constant composite patch modulus in a longitudinal direction ($E_{2x}^{ref}$ = 626 MPa) and a constant cover patch and a parent belt Young’s modulus. At first, the decrease of the Young’s modulus in a longitudinal direction led to a slight decrease in the maximum shear stress of interface (2) but an increase in the shear stress in interface (1), as shown in Figure 18a. The intersection of these two curves in the intersection point value of $E_{2x}=19\;\mathrm{MPa}$ should give the optimum shear stress distribution. The distribution of the maximum normal stress in the parent belt and the inner patch had a significant influence with the variation of the Young’s modulus in a longitudinal direction, as shown in Figure 19a. The optimum normal stress distribution was obtained with the decrease of (1/5) of the reference longitudinal Young’s modulus. As a result, the reducing of fiber rigidity in the load direction gave the optimum system performance but the fiber rigidity in a transverse direction did not have a significant influence on the performance of the repaired system. For the conveyer belt, it was better to use the stacking sequence [90]_2_ for the reparation of the specimen.

## 5. Summary

In order to be able to repair a composite conveyor belt by bonding technology, a bidirectional linear analysis was performed for repair stress concentration prediction. Only the shear stress inside the adhesive was taken into account while normal stress occurred in the parent belt, the inner and the cover patch. A repaired belt system was exposed to hygro-thermomechanical loads. The successful method of repair optimization was to balance the stress distribution in the rubber to the composite patch and in the adhesive interfaces. Experimental observations together with a theoretical analysis have shown that damage occurred in the stress concentration zone and a compressive normal stress was produced in the load direction at the cover patch so the damage initiation started on the longitudinal edges of the covered patch. The fibers of the inner patch were then broken by means of the tensile stress concentration located near the longitudinal and transverse edges of the inner patch. Finally, the destruction of the parent belt was exhibited at the center. As the experimental results proved, the cover patch was still attached to the parent belt at the transverse edge even if the fibers were broken and completely separated from the parent belt. The cover patch was particularly damaged and the hyper-elastic behavior led to the absorption of the transfer shear stress of the most contracted interface, which could be found between the inner and cover patches.

The effects of Young’s modulus and the thickness of the adhesive and the rigidity of the composite patch on the performance of the repair system were examined by means of a parametric study. This model could be generalized to the repair bonding of rubber to rubber and rubber to composite internal or/and external patch applications.

The hot vulcanization method is demonstrated in [14] and a comparison between hot and cold joining is well investigated.

## Figures and Tables

**Figure 1 polymers-13-01710-f001:**
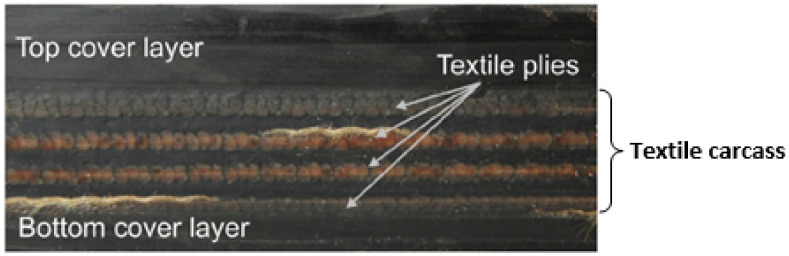
Rubber-textile composite conveyor belt.

**Figure 2 polymers-13-01710-f002:**
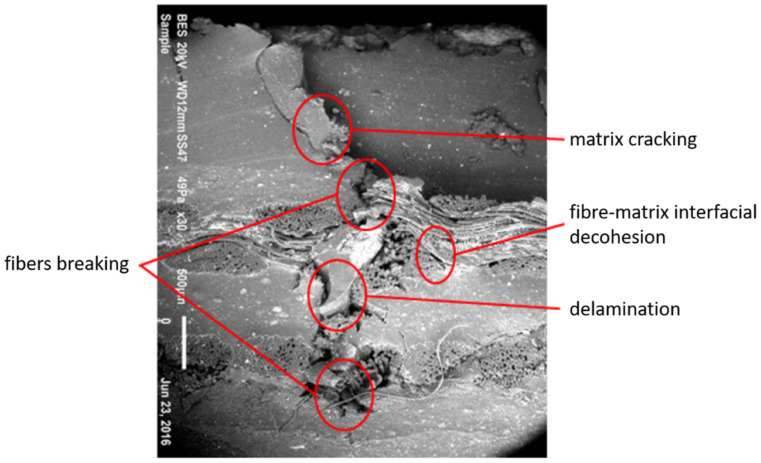
Micrograph of the cross-sectional area of the impacted specimen at 48.51 Joule.

**Figure 3 polymers-13-01710-f003:**
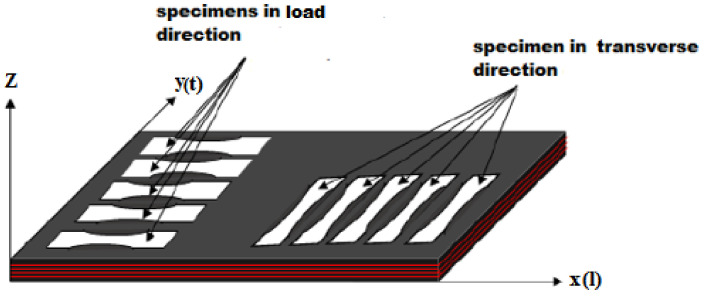
Belt view of the tensile specimen distribution with respect to longitudinal and transverse directions.

**Figure 4 polymers-13-01710-f004:**
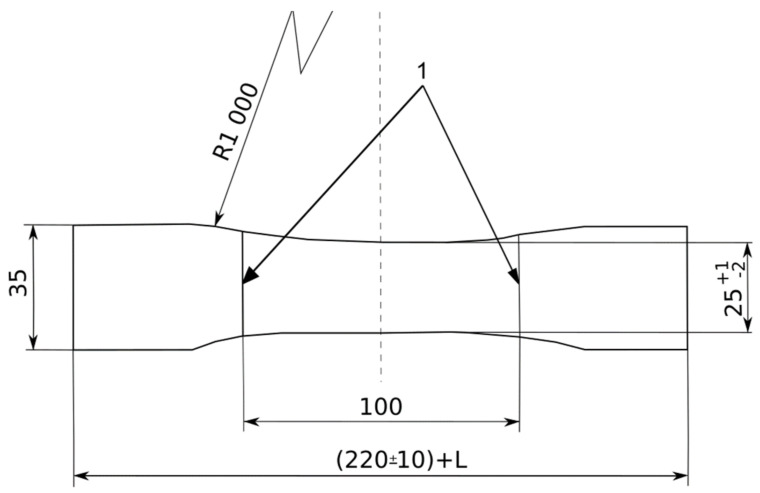
Standardized belt specimen for the uniaxial tensile test according to ISO 283.

**Figure 5 polymers-13-01710-f005:**
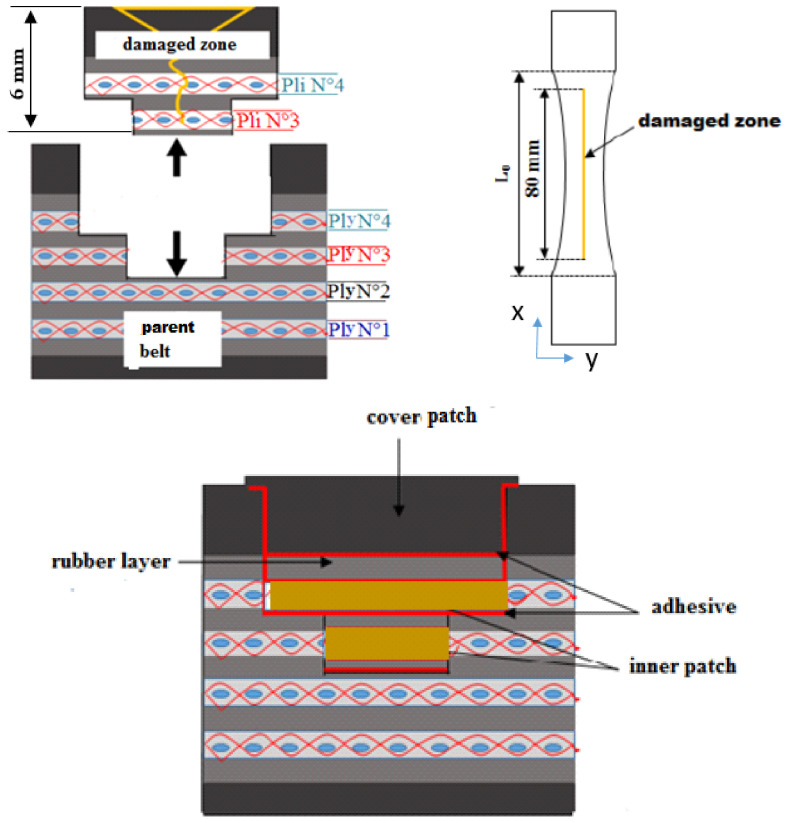
Diagram of the bonded belt patch repair technology specimen.

**Figure 6 polymers-13-01710-f006:**
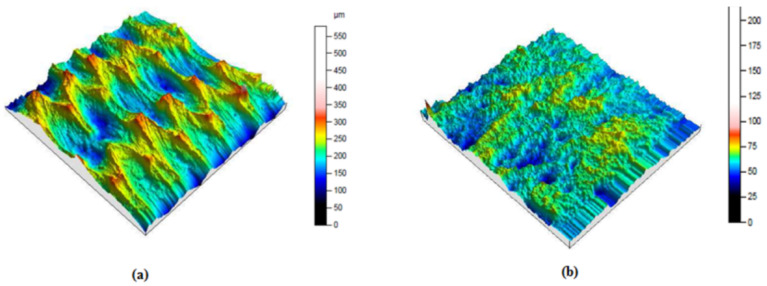
Topography of the processed strip sample on 36 mm^2^ with a path of 20 mm. (**a**) by a peeling process, Ra = 13.8 mm (**b**) by a milling process, Ra = 7.76 mm.

**Figure 7 polymers-13-01710-f007:**
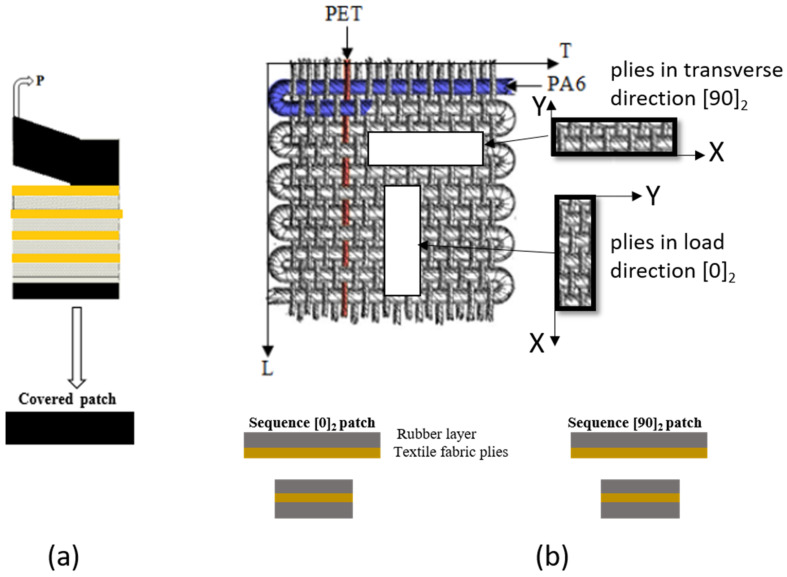
Preparation of the inner and covered patch: (**a**) peel technique; (**b**) cutting fiber textile ply.

**Figure 8 polymers-13-01710-f008:**
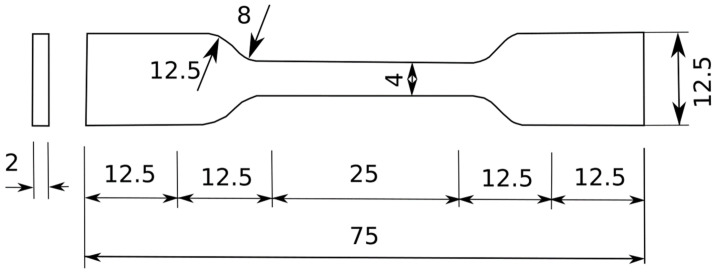
Standardized adhesive specimen for the uniaxial tensile test according to ISO37: 2011.

**Figure 9 polymers-13-01710-f009:**
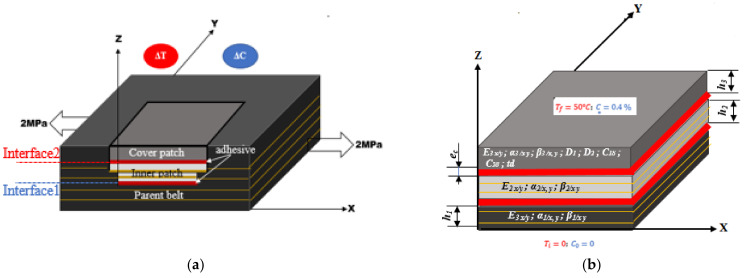
Schematic representation of the belt repaired composite structure under mechanical tension in the presence of the temperature gradient ∆T and the humidity concentration ∆C: (**a**) transverse section with two interfaces; (**b**) geometric parameters and material properties needed for the 2D model.

**Figure 10 polymers-13-01710-f010:**
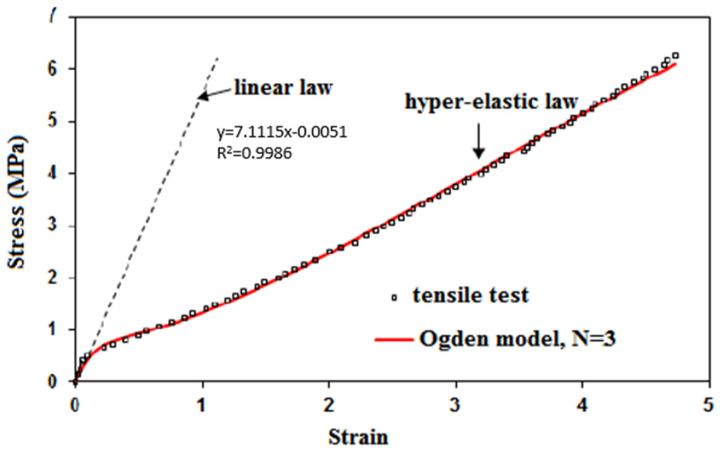
Tensile behavior of cold polymerized adhesive ELASTOGLUE 2000.

**Figure 11 polymers-13-01710-f011:**
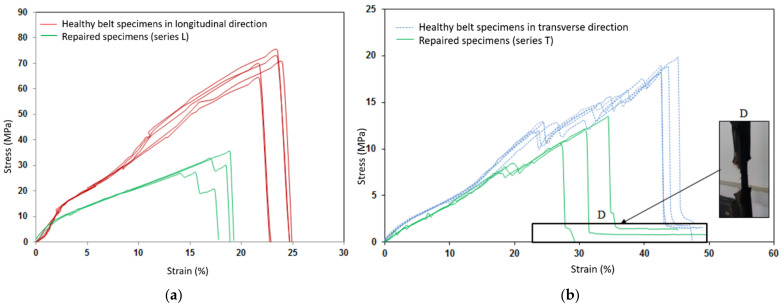
Comparison of the tensile behavior of the heathy and repaired specimens by the cold polymerization process: (**a**) in a longitudinal direction (**b**) in a transverse direction.

**Figure 12 polymers-13-01710-f012:**
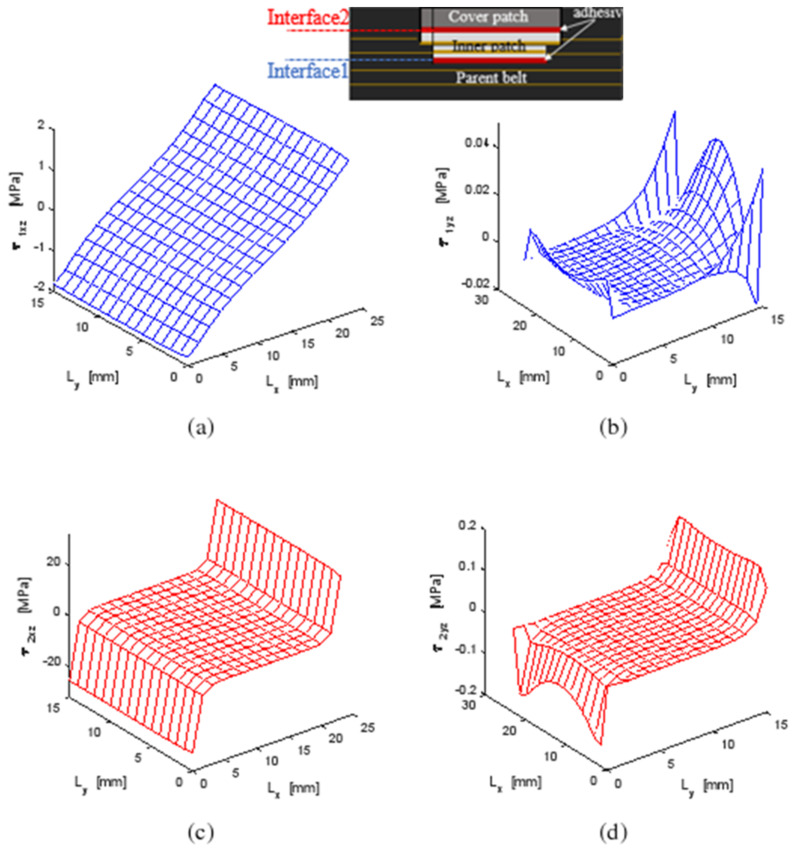
Shear stress distribution under a hygro-thermomechanical load on XZ and YZ planes in (**a**,**b**) Interface (1); (**c**,**d**) Interface (2).

**Figure 13 polymers-13-01710-f013:**
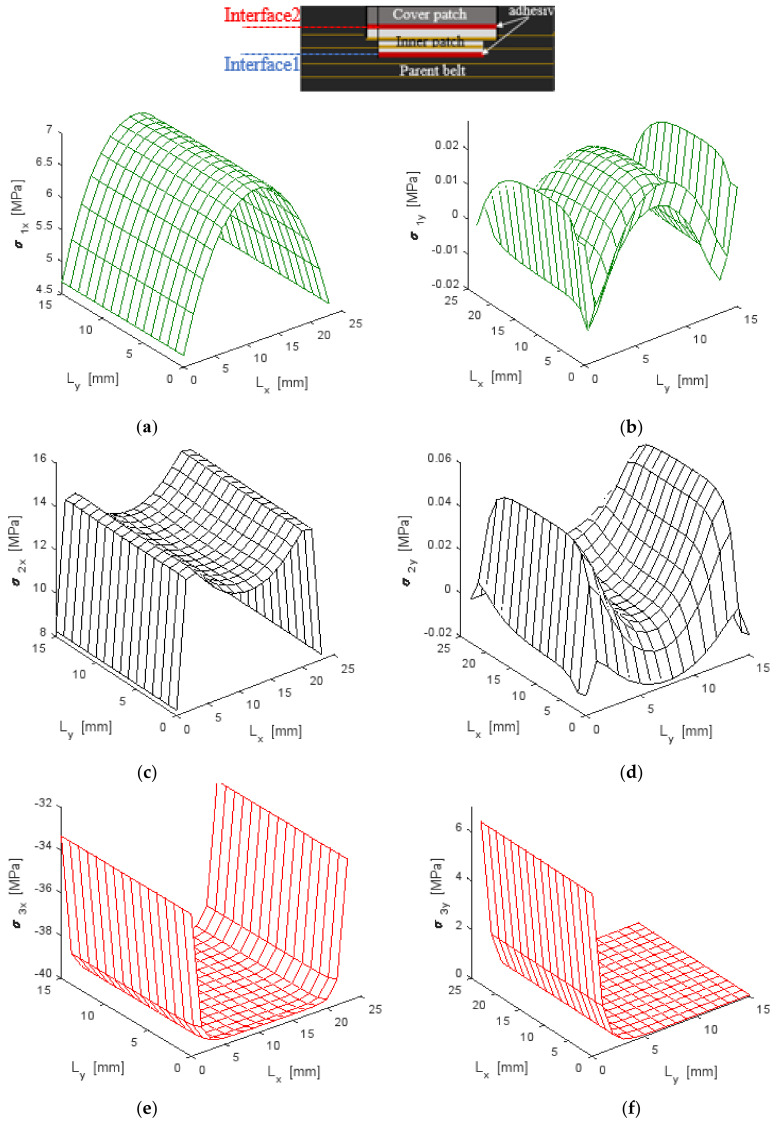
Normal stress distribution under a hygro-thermomechanical load (2 MPa, 50 °C and 40% RH) in longitudinal and transverse directions in (**a**,**b**) a parent belt; (**c**,**d**) an inner patch; (**e**,**f**) a cover patch.

**Figure 14 polymers-13-01710-f014:**
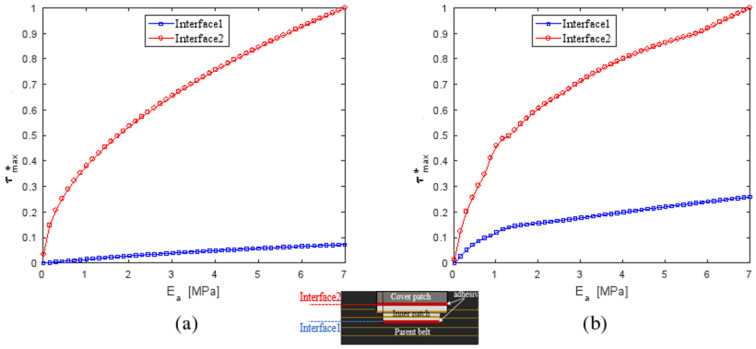
Variation of the maximal normalized shear stress ratio $\tau_{max}^{*}$ in Interface (1) and Interface (2) as a function of the adhesive Young’s modulus E_a_ in: (**a**) XZ plane; (**b**) YZ plane.

**Figure 15 polymers-13-01710-f015:**
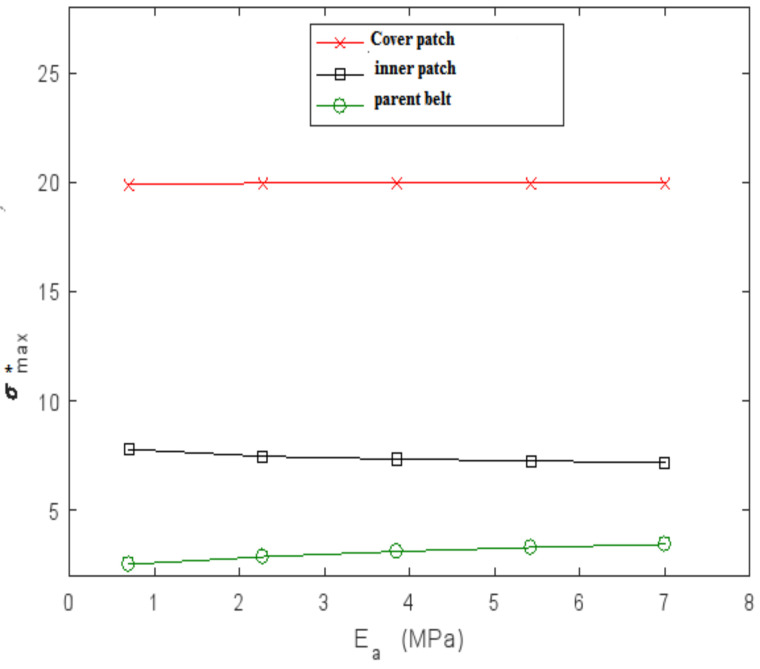
Variation of the maximal normalized normal stress ratio $\sigma_{max}^{*}$ in adherents as a function of adhesive Young’s modulus E_a_.

**Figure 16 polymers-13-01710-f016:**
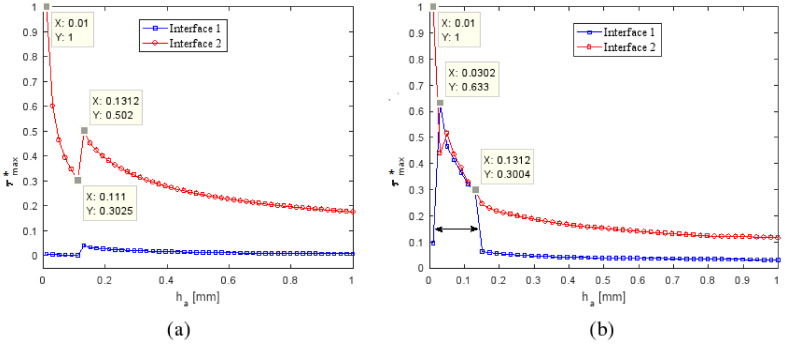
Variation of the maximal normalized shear stress ratio $\tau_{max}^{*}$ in interface (1) and interface (2) as a function of adhesive thickness *h_a_* in: (**a**) XZ plane; (**b**) YZ plane.

**Figure 17 polymers-13-01710-f017:**
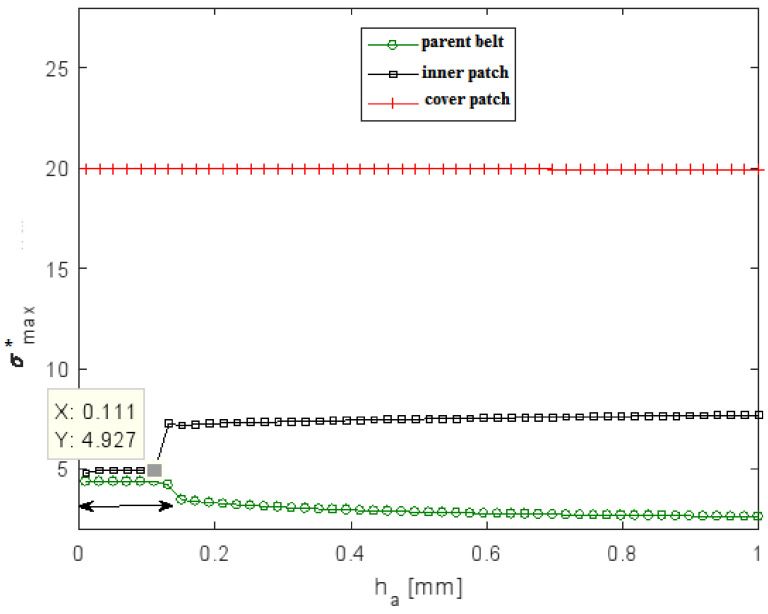
Variation of the maximal normalized normal stress ratio $\sigma_{max}^{*}$ in adherents as a function of adhesive thickness *h_a_*.

**Figure 18 polymers-13-01710-f018:**
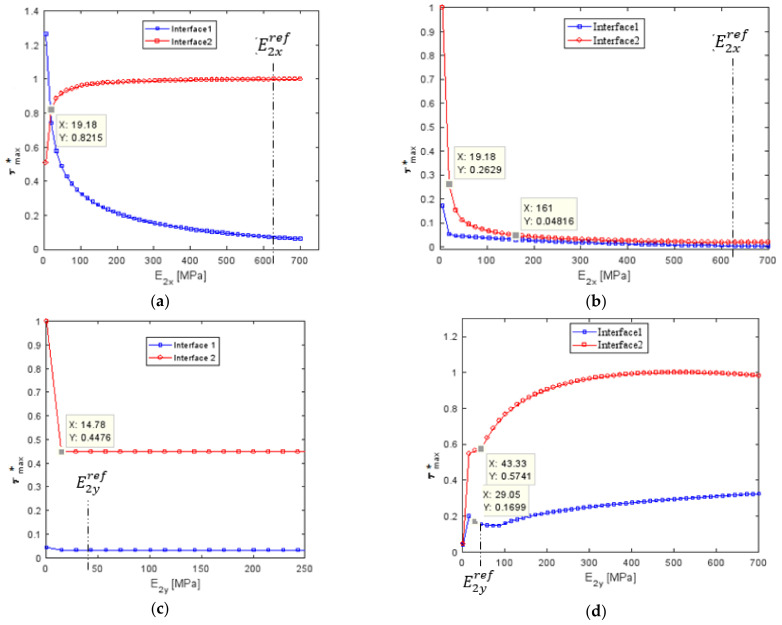
Variation of the maximal normalized shear stress in XZ-YZ plane as a function of: (**a**,**b**) longitudinal Young’s modulus; (**c**,**d**) transverse Young’s modulus.

**Figure 19 polymers-13-01710-f019:**
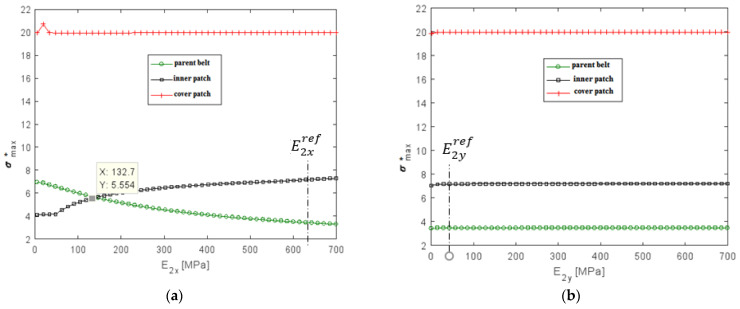
Variation of the maximal normalized normal stress as a function of (**a**) longitudinal Young’s modulus; (**b**) transverse Young’s modulus.

**Table 1 polymers-13-01710-t001:** Mechanical properties of materials.

	Materials	Belt	Textile Carcass PET/Pa6	Cover Rubber	ELASTOGLUE 2000
Properties					
E_x_ (MPa)		550.77	626.05	3.42	7.11
E_y_, E_z_ (MPa)		41.85	46.54	-	-
ν_xy_		0.38	0.38	0.33	0.4
R_e,x_ (MPa)		10	10.12	-	-
R_e,y_ (MPa)		1.29	1.34	-	-

**Table 2 polymers-13-01710-t002:** Geometric properties of the repaired belt.

Parameters	Dimension [mm]
Overlap length *L_x_*	25
Overlap width *L_y_*	15
Belt thickness *e_b_*	10
Covered patch thickness *h*_3_	3
Inner patch thickness *h*_2_	2.42
Adhesive thickness *e_c_*	0.15
Parent belt thickness *h*_1_	4.27

**Table 3 polymers-13-01710-t003:** Coefficients of thermal expansion (CTE) of the assembly constitutive materials.

Materials	CTE (×10^−6^)
Polyester (PET)	25
Polyamide (Pa6)	15
Rubber	10

**Table 4 polymers-13-01710-t004:** Moisture diffusion sequential dual Fickian (SDF) law coefficients of the covering rubber.

Model	*D*_1_ (× 10^−11^ m^2^/s)	*D*_2_ (× 10^−14^ m^2^/s)	*C*_1∞_ = *C*_∞R_ (%)	*C*_∞_ (%)	$\sqrt{t_{d}}$ (*s*^1/2^)
SDF	2.14	5.25	5.32	37.89	380

**Table 5 polymers-13-01710-t005:** Hygric expansion coefficients of materials under water immersion at 50 °C.

Materials	β (%−1)
Upper covering rubber	0.00530
Belt in the longitudinal direction (PET)	0.00265
Carcass in the longitudinal direction (PET)	0.00165
Carcass in the transverse direction (PA6)	0.00136

**Table 6 polymers-13-01710-t006:** Repairing performance for both longitudinal and transverse specimens.

Specimens	Patch Stacking Sequence	Young’s Modulus $E^{rep}$ (MPa)	Failure Resistance $R^{rep}$ (MPa)	$\frac{R^{rep}}{R^{ref}}$ (%)	$\frac{E^{rep}}{E^{ref}}$ (%)
L-1	[0]_2_	440.9	33.96	48.53	80.05
L-2		439.4	24.96	35.66	79.77
L-3		439.47	31.3	44.71	79.79
Average for L		439.92	30.07	42.96	79.87
T-1	[90]_2_	34.89	9.77	52.07	83.35
T-2		37.84	11.18	59.59	90.4
T-3		36.34	12.9	68.73	86.82
Average for T		36.35	11.28	60.13	86.86

## Data Availability

The data presented in this study are available on request from the corresponding author.

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
