# Peer review of "Theoretical and Experimental Investigation of Bonded Patch Repairs of a Rubber Reinforced Composite Conveyor Belt"

_polymers, 2021, doi:10.3390/polym13111710_

Round 1

Reviewer 1 Report

Dear Authors.

I have read your study carefully and it seems interesting. It is extensive and quite comprehensive. It contains calculations validated in experiments, resulting in development of universal mathematical model. However, I do not think that the obtained model will be applied by teams repairing conveyor belts (it is strongly technical activity), it does not detract from the scientific value of the procedure presented, which can also be used in other applications. Therefore, I admit the publication of the study in “Polymers”. Nevertheless, I propose taking into account minor remarks given below.

  1. Lack of keywords.

  1. English should be carefully checked and corrected within whole manuscript! Examples of errors:

- Page 3, line 85: “This damage present the critical case...” – should be “presents”.

- Page 7, line 176/177: “their mechanical characteristics does not depend” – should be “do not”.

- Page 9, line 230: I think you ment “healthy specimen” instead of “health specimens”. It refers to whole text – the “health specimen” is used many times.

- Page 17, line 375: I think that “composite conveyor belt” would sounds better than “conveyor composite belt”.

  1. In my opinion “Summary” would be better instead of “Conclusion” for the applied form of expression.

Sincerely

Author Response

  

1.Lack of keywords. 

 Rubber, textile reinforcement, experimental results, bidirectional linear analysis, hygro-thermo mechanical loading, bonding assembly 

2.English should be carefully checked and corrected within whole manuscript! Examples of errors: 

- Page 3, line 85: “This damage present the critical case...” – should be “presents”. 

This damage presents the critical case 

- Page 7, line 176/177: “their mechanical characteristics does not depend” – should be “do not”. 

their mechanical characteristics do not depend 

- Page 9, line 230: I think you ment “healthy specimen” instead of “health specimens”. It refers to whole text – the “health specimen” is used many times. 

Page6, line 149; healthy belt 

Page9, line 226; same behavior as those on the healthy belt specimens 

Page9, line 230; Failure resistance of healthy specimen. 

Page9, line 230; same behavior as those on the healthy specimens 

- Page 17, line 375: I think that “composite conveyor belt” would sounds better than “conveyor composite belt”. 

 Page1, line29/30, composite conveyor belt 

Page17, line375, composite conveyor belt 

3.In my opinion “Summary” would be better instead of “Conclusion” for the applied form of expression. 

It is done 

Reviewer 2 Report

The authors proposed a reparation method for designing and optimizing a rubber to rubber and rubber to textile reinforcement for conveyor belt used in transport industry. A bidirectional linear analysis was conducted to predict the geometric parameters effect on the stress concentration zone of the repaired belt under hygro-thermo mechanical loading and its consequence on the integrity of the structure. Additionally, tensile test was carried out to investigate the behavior of repaired specimen made with rubber cover patch and inner composite patch. The results showed a good correlation between theoretical and experimental data, providing a strength tool to understand the load transfer from rubber to textile rubber patch. The work is interesting and can be published in Polymers if the following issues can be addressed:

  1. The authors should cite the paper “Nanocomposites for electronic applications that can be embedded for textiles and wearables” (Science China Technological Sciences, 2019, 62, 895-902) and “Strong, lightweight, and highly conductive CNT/Au/Cu wires from sputtering and electroplating methods” (Journal of Materials Science & Technology, 2020, 40, 99-106) in the introduction section for better review of fiber reinforced composite.
  2. Figure 1 quality is quite low. Can the authors provide better quality image?
  3. In Figure 11b, why did the stress-strain curves of the repaired specimens have D region while no similar regions were found in the other curves?
  4. In Figure 16, why did the shear stress suddenly increase when thickness increased from 0.11 to 0.13 for XZ plan and from 0.03 to 0.13 for YZ plan?
  5. Several typos were found in the manuscript (“rapidity o process”, “Khalili and al.”…). The authors should correct them. 

Author Response

  1. The authors should cite the paper “Nanocomposites for electronic applications that can be embedded for textiles and wearables” (Science China Technological Sciences, 2019, 62, 895-902) and “Strong, lightweight, and highly conductive CNT/Au/Cu wires from sputtering and electroplating methods” (Journal of Materials Science & Technology, 2020, 40, 99-106) in the introduction section for better review of fiber reinforced composite.

It is done 

  1. Figure 1 quality is quite low. Can the authors provide better quality image?

The Figure 1 is changed 

  1. In Figure 11b, why did the stress-strain curves of the repaired specimens have D region while no similar regions were found in the other curves?

The D region refer to the covered patch attachment, it is observed that the covered patch is maintained only for the transverse specimens, this observation proves that the orientation of the specimens relative to the stacking sequence and the rigidity of the inner patch have a significant effect on the rupture mechanism and the mode of repaired specimen damage. 

  1. In Figure 16, why did the shear stress suddenly increase when thickness increased from 0.11 to 0.13 for XZ plan and from 0.03 to 0.13 for YZ plan?

In fact, the shear mechanism of the adhesive layer becomes more significant when thickness increased from 0.11 to 0.13 for XZ plan and from 0.03 to 0.13 for YZ plan,  

  1. Several typos were found in the manuscript (“rapidity o process”, “Khalili and al.”…). The authors should correct them.

It is done